# Laparoscopic Adenomyomectomy under Real-Time Intraoperative Ultrasound Elastography Guidance: A Case Series and Feasibility Study

**DOI:** 10.3390/jcm11226707

**Published:** 2022-11-12

**Authors:** Yoshiaki Ota, Kuniaki Ota, Toshifumi Takahashi, Yumiko Morimoto, Soichiro Suzuki, Rikiya Sano, Mitsuru Shiota

**Affiliations:** 1Department of Gynecologic Oncology, Kawasaki Medical School, Okayama 701-0192, Japan; 2Fukushima Medical Center for Children and Women, Fukushima Medical University, Fukushima 960-1295, Japan

**Keywords:** adenomyomectomy, adenomyosis, elastography, laparoscopic surgery

## Abstract

Background: This study aimed to examine the clinical characteristics of 11 patients undergoing laparoscopic adenomyomectomy guided by intraoperative ultrasound elastography and this technique’s feasibility. Patients and Methods: Eleven patients undergoing laparoscopic adenomyomectomy using ultrasound elastography for adenomyosis at Kawasaki Medical School Hospital in Okayama, Japan between March 2020 and February 2021 were enrolled. Operative outcomes included operative time, operative bleeding, resected weight, operation complications, percent change in hemoglobin (Hb) values, and uterine volume pre- and postoperatively. Dysmenorrhea improvement was evaluated by changes in visual analog scale (VAS) scores pre- and 6- and 12-months postoperatively. Results: The median operative time and bleeding volume was 125 min (range, 88–188 min) and 150 mL (10–450 mL), respectively. The median resected weight was 5.0 g (1.5–180 g). No intraoperative or postoperative blood transfusions or perioperative complications were observed. The median changes in uterine volume, Hb value, and VAS score were −49% (−65 to −28%), −3% (−11 to 35%), and −80% (−100 to −50%), respectively. The median follow-up period post-surgery was 14 months (7–30 months). Adenomyosis recurrence was not observed in the patients during the follow-up period. Conclusions: Laparoscopic adenomyomectomy using ultrasound elastography guidance is minimally invasive and resects as many adenomyotic lesions as possible.

## 1. Introduction

Adenomyosis is a benign disorder characterized by ectopic endometrial glands and stroma within the myometrium [1]. Ectopic endometrial tissue induces hypertrophy and hyperplasia of the surrounding myometrium, resulting in a diffusely enlarged uterus [2]. These abnormal modifications greatly impact the quality of life of women of reproductive age, leading to dysmenorrhea, pelvic pain, abnormal uterine bleeding, and infertility [3].

Adenomyosis treatment primarily depends on the severity of symptoms and reproductive circumstances [4]. Hysterectomy is the conventional treatment for women with symptomatic adenomyosis. However, some patients, especially women of reproductive age with severe symptoms that are unresponsive to pharmacotherapy and have failure with infertility treatment, may be candidates for adenomyomectomy. Unlike fibroids, adenomyosis does not have a well-defined border; therefore, complete enucleation may be difficult.

Ultrasound elastography is a non-invasive imaging tool that delivers information regarding lesions and surrounding tissue stiffness to diagnose mammary neoplasia [5] and stage liver fibrosis [6]; it has emerged as a valuable addition to conventional transvaginal ultrasound in increasing the diagnostic accuracy of adenomyosis [7,8]. Additionally, intraoperative elastography has been used to detect focal lesions in breast and brain tumors [5,9]. One of our recent studies reported successful laparoscopic excision of an adenomyotic lesion, using intraoperative ultrasound elastography [10,11].

This study aimed to examine the clinical characteristics of 11 patients undergoing laparoscopic adenomyomectomy guided by intraoperative ultrasound elastography and the feasibility of this technique. The novelty of this technique is the intraoperative monitoring of the adenomyosis site using ultrasonic elastography while resecting as many adenomyotic lesions as possible.

## 2. Patients and Methods

### 2.1. Study Design

This study was a case series study with no control group.

### 2.2. Ethical Approval

The study protocol was approved by the ethics committee of Kawasaki Medical School (approval no. 5075-00), and all procedures involving human participants followed the ethical standards of the institutional review board of the study center.

### 2.3. Patients

Eleven patients who underwent laparoscopic adenomyomectomy using ultrasonic elastography for adenomyosis at Kawasaki Medical School between March 2020 and February 2021 were enrolled in this study. These were patients with hypermenorrhea or dysmenorrhea who wished to preserve their uterus, whose symptoms could not be adequately controlled with hormone therapy, or who wished to have a baby. All patients received a written explanation of the surgery and provided their informed consent.

### 2.4. Diagnosis of Adenomyosis and Calculation of Uterus and Adenomyosis Lesion Volume

Screening for adenomyosis was performed using transvaginal ultrasonography and the final diagnosis was made using magnetic resonance imaging (MRI) [12]. This study included patients with tumor-diffuse-forming intrinsic or extrinsic adenomyosis. However, patients with extensive adenomyosis were not considered for surgery [13] (Figure 1). Uterus and adenomyotic lesions were treated as triaxial ellipsoids; their volumes were calculated using the following formula: volume = 0.523 × longitudinal diameter × anteroposterior diameter × transverse diameter [14].

### 2.5. Pre- and Postoperative Hormonal Treatment and Evaluation of Dysmenorrhea

Some patients were administered oral dienogest (2 mg/day) as preoperative or postoperative hormone therapy. No other hormonal treatment, such as oral contraceptives or levonorgestrel-releasing intrauterine system, was administered. Dysmenorrhea was rated on a 10-point scale using a visual analog scale (VAS) before and after surgery [15].

### 2.6. Surgical Procedures

One gynecologic surgeon (YO) performed all the surgeries. Surgical procedures for laparoscopic adenomyomectomy under ultrasonic elastography guidance have been described in our previous reports [10,11]. A 10-mm port (ENDOPATH XCEL^®^, Ethicon Endo-Surgery, Cincinnati, OH, USA) for the zero-degree laparoscope was introduced intra-umbilically, and three additional 5-mm lateral ports (ENDOPATH XCEL^®^, Ethicon Endo-Surgery) were placed under vision, centrally, and on the left and right lateral sides. The surgeon used central and left-sided lateral ports to perform most surgical procedures. Vasopressin (diluted to 1 IU/70 mL normal saline) was injected into the uterine wall to decrease intraoperative bleeding. Eleven surgical scalpels (Feather Safety Razor Co. Ltd., Tokyo, Japan) were inserted from the superior margin of the pubis symphysis into the abdominal cavity and a longitudinal incision was made in the uterine fundus (Figure 2a). The adenomyotic lesions were carefully excised with a surgical scalpel along the anterior wall of the uterus (Figure 2b). After the initial enucleation of the adenomyotic tissues, intraoperative real-time tissue elastography (ARIETTA 850, Hitachi, Ltd., Tokyo, Japan) revealed residual adenomyosis in a small part of the uterine wall, depicted as bright blue areas (Figure 2c). The residual adenomyotic tissues were completely resected using scissor forceps (Figure 2d) until bright blue areas were converted to detect green areas which indicated the normal myometrial tissue on real-time elastography (Figure 2e). After enucleating the adenomyotic tissues maximally, the uterine incisions were repaired in multiple layers using barbed sutures (0 Stratafix^®^ Symmetric PDS^®^ Plus, Ethicon Endo-Surgery) [16] (Figure 2f). Myomectomy and cauterization of the endometriotic lesions were performed when coexisting uterine fibroids and endometriosis were observed. Finally, the suture layer was covered with an anti-adhesion sheet (INTERCEED^®^; Ethicon Endo-Surgery). The resected lesions, diagnosed as adenomyosis using MRI, were histopathologically confirmed as adenomyosis.

### 2.7. Operative Outcome Measures

Objective operative outcomes included operative time, operative bleeding, resected weight, operation complications, percent changes in hemoglobin (Hb) values, and uterine volume between pre- and postoperatively. Intraoperative bleeding volume was defined as the gauze weight plus the difference between the volume of saline delivered and the volume of aspirated saline used for cleaning. Surgical complications were evaluated according to the Clavien–Dindo classification of surgical complications modified by Katayama et al. [17]. Postoperative Hb levels were measured 1 month after surgery. Postoperative changes in uterine volume were evaluated using MRI before and 1 month after surgery. The improvement in dysmenorrhea was evaluated by the changes in VAS scores pre- and 6- and 12-months postoperatively.

### 2.8. Postoperative Follow-Up and an Evaluation of Recurrence

After the postoperative visit in the first month, all patients underwent transvaginal ultrasound and clinical examinations for evaluating recurrence of adenomyosis every 3 months. Screening for recurrence of adenomyosis was performed by transvaginal ultrasonography, and in cases where thickening of the myometrium was observed, MRI was performed to diagnose recurrence [12].

### 2.9. Statistical Analysis

All statistical analyses were performed with EZR (Saitama Medical Center, Jichi Medical University, Saitama, Japan), which is a graphical user interface for R (The R Foundation for Statistical Computing, Vienna, Austria) [18]. The Wilcoxon signed-rank test was used for paired comparisons of each parameter before and after the surgery. Statistical significance was considered significant when the *p*-value was < 0.05.

## 3. Results

### 3.1. Preoperative Patient Characteristics

Preoperative clinical characteristics of the 11 patients are presented in Table 1. The median age of the patients was 35 years (range: 31–42 years). None of the patients had any history of delivery. The chief complaints were as follows: four patients had only hypermenorrhea, five patients had only dysmenorrhea, one patient had both hypermenorrhea and dysmenorrhea, and one patient wanted to have a child. The adenomyosis lesion was located on the posterior wall in seven patients and on the anterior wall in four patients. The median uterine and adenomyosis lesion volumes were 224 cm^3^ (range, 78–518 cm^3^) and 58 cm^3^ (range, 11–155 cm^3^), respectively. The median value of Hb was 12.5 g/dL (range, 7.8–14.9 g/dL). The median VAS score was 8 (range: 3–10). Nine patients (81.8%) had been taking dienogest as hormone therapy (HT) before surgery, although HT did not improve their symptoms.

### 3.2. Operative Outcomes

The operative outcomes are summarized in Table 2. Laparoscopic adenomyomectomy alone was performed in five cases, and concurrent myomectomy and removal of endometriosis lesions, such as endometriotic cysts and deep endometriosis, were performed in six cases. The median operative time and bleeding volume was 125 min (range, 88–188 min) and 150 mL (10–450 mL), respectively. There was not a significant difference in operative time and bleeding volume between only adenomyotic lesion and adenomyosis co-existing with other tumors (*p* = 0.08, *p* = 0.47, respectively) although adenomyosis often coexists with uterine fibroids, deep endometriosis, and ovarian endometrioma. The median resected weight was 5.0 g (range, 1.5–180 g). After surgery, the median uterine volume was 104 cm^3^ (range 45–261 cm^3^). No intraoperative or postoperative blood transfusions or perioperative complications were observed in all 11 patients. The median follow-up period after surgery was 14 months (range: 7–30 months). The median Hb level after surgery was 12.5 g/dL (range 10.5–13.9 g/dL). The median VAS scores 6 and 12 months after surgery were 2 (range, 0–5) and 0 (range, 0–5), respectively. There was no recurrence of adenomyosis in the patients who underwent laparoscopic adenomyomectomy using ultrasonic elastography guidance during the follow-up period. Table 3 and Figure 3 summarize the changes in uterine volume, Hb level, and VAS score before and after surgery. The median changes in uterine volume, Hb value, and VAS score were, −49% (range, −65 to −28%), −3% (−11 to 35%), and −80% (−100 to −50%), respectively. In individual patients, uterine volume and VAS score, a measure of dysmenorrhea, decreased significantly after surgery; however, the Hb level did not change after surgery. Nine patients (81.8%) hoped to take dienogest to a prevent postoperative recurrence, while one patient (9.1%) did not hope to take it because of their desire to have a baby, and another patient (9.1%) refused to take it due to the side effect with nausea (Figure 3).

## 4. Discussion

This study described the clinical course of 11 women with hormone therapy-refraction or infertility who underwent laparoscopic adenomyomectomy and wished to preserve their uterus. Eleven patients completed the procedure reasonably, with no postoperative complications and no recurrence during the follow-up period.

Complete surgical removal of uterine adenomyotic lesions is difficult. Adenomyosis is characterized by endometrial glands and interstitial invasion of the myometrium. Therefore, when adenomyotic tissue is removed, the uterine wall is removed together with the lesion [19]. Furthermore, the boundaries of adenomyotic lesions are often unclear. Residual adenomyosis lesions are inevitable even when resected using gross color differences as landmarks. Therefore, we decided to perform resection using intraoperative ultrasound, which is used in myomectomy of uterine fibroids [20]. Since it is difficult to distinguish the normal muscle layer from the adenomyosis area using conventional ultrasonography, we proposed the use of ultrasonic elastography [10,11].

Elastography is a new imaging technique used to detect adenomyosis, quantify the local mechanical properties of the tissue, and assess its stiffness. There are differences in tissue stiffness of the normal myometrium, uterine fibroids, and adenomyosis. Therefore, recent studies have reported the use of this imaging technique to diagnose adenomyosis by detecting the characteristic tissue hardness. Zhang et al. reported the possibility of detecting adenomyosis using elastography based on the global increase in the hardness of the myometrium caused by adenomyosis [7]. Stoelinga et al. stated that the definitive diagnosis of adenomyosis using ultrasound elastography correlates with histology and MRI [21]. These previous studies have confirmed that it is possible to identify residual adenomyosis lesions intraoperatively.

In addition to resecting the adenomyotic lesion, adenomyomectomy involves a suture method to maintain uterine strength. Osada developed the triple-flap technique for adenomyomectomy [22,23], which was validated in 104 patients [23]. Of the 26 patients who underwent adenomyomectomy, 16 could conceive, and 14 patients went on to have deliveries. However, in general, uterine rupture associated with pregnancy after enucleation of adenomyosis uteri is reported to be 2.8% per case and 2.3% per pregnancy [22], and no uterine rupture was observed in patients who underwent adenomyomectomy using the triple flap technique [23]. In laparoscopic surgery, it is difficult to maintain constant tension on the muscle layer during suturing, and there is often concern that incomplete muscle layer repair may occur because of suture loosening. Therefore, we performed continuous sutures using barbed threads to increase suture strength at the site of adenomyomectomy because barbed threads do not require ligation and do not loosen after suturing [16].

To date, adenomyomectomy has been performed under laparotomic, laparoscopic-assisted, and laparoscopic surgery. Nishida et al. reported the surgical results and postoperative symptom recurrence in laparotomic adenomyomectomy [24]. In that report, surgery was performed in 44 patients, with a mean operating time of 159 min (range, 107–305 min), mean blood loss of 745 g (range, 50–2951 g), and mean resected mass weight of 281 g (range, 46–1300 g). The degree of dysmenorrhea before and after surgery decreased from a mean preoperative VAS score of 9.4 (range, 6.5–10) to 0.8 (range, 0–4.0). Approximately 1 year after the surgery, 32 patients (73%) experienced symptom relapse. Ahn et al. compared the operative feasibility of laparoscopic laparoscopy-assisted and adenomyomectomy. Laparoscopic adenomyomectomy was performed in 82 patients and bilateral uterine artery ligation or temporary uterine arterial occlusion was performed in 75 patients. The mean operative time, blood loss, and resected mass weight were 99.9 min (range, 60–225), 119.5 g (range, 30.0–500.0), and 39.3 g (2.0–150.0), respectively, and there were no blood transfusions during the operation. This study did not mention postoperative recurrence. Taken together, the operative outcomes of these reports were comparable to those of our laparoscopic adenomyomectomy under ultrasonic elastography guidance.

The feature of the current study is that adenomyomectomy was performed for patients with dienogest-resistant adenomyosis, and postoperative dienogest was significantly effective. Previously, we reported that taking dienogest after microwave endometrial ablation to shrink the adenomyotic lesion was efficient in a patient with dysmenorrhea refractory to dienogest [25], and most patients had a similar clinical course in the current series. It may be supposed that the precise reduction of adenomyotic lesions by laparoscopic adenomyomectomy under ultrasonic elastography guidance may have led to the drug effect of dienogest.

The present study has several limitations. The fact that no recurrence of adenomyosis or dysmenorrhea was observed in 11 patients does not rule out their occurrence in the future, as the observation period was short in our study. Further research is needed to determine the risk of long-term recurrence and uterine rupture during pregnancy and to analyze whether this technique can contribute to improve the fecundity as fertility-sparing surgery or not. Moreover, our report did not examine the benefits of this technique over other techniques, and further evaluation of this technique at different settings may be necessary.

## 5. Conclusions

Laparoscopic adenomyomectomy using ultrasound elastography guidance is minimally invasive and resects as many adenomyotic lesions as possible. This technique is comparable to the techniques reported in previous reports on adenomyomectomy and is highly feasible in real-world clinical practice.

## Figures and Tables

**Figure 1 jcm-11-06707-f001:**
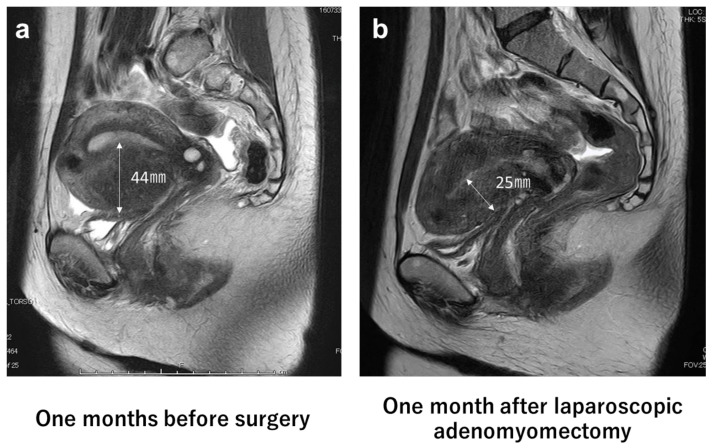
Magnetic resonance imaging (MRI) of pre- and post-laparoscopic adenomyomectomy. (**a**) photograph of sagittal T2-weighted pelvic MRI 1 month before surgery. The image shows substantial adenomyosis in the anterior myometrium (44 mm); (**b**) photograph of sagittal T2-weighted pelvic MRI 1 month after laparoscopic adenomyomectomy. The image shows no residual adenomyosis and a decrease in the thickness of the myometrium (25 mm).

**Figure 2 jcm-11-06707-f002:**
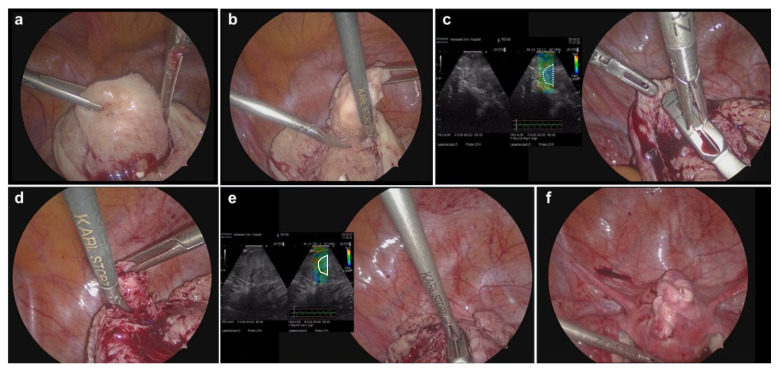
Representative photographs of laparoscopic adenomyomectomy under elastography guidance. (**a**) An adenomyosis lesion on the anterior wall was excised with a scalpel to preserve the serosa. (**b**) During the excision of the adenomyosis lesion on the anterior wall, scissors forceps were used in areas that could not be excised with a scalpel due to angulation. (**c**) After resection of the adenomyosis lesion, elastography was used to confirm the residual disease, with areas appearing in blue (white dotted circle) as harder tissue than the surrounding area, which was considered to have residual lesions. (**d**) The residual lesion was excised using scissors forceps. (**e**) Real-time tissue elastography confirmed complete resection without residual lesions, which appear as normal myometrial tissue in green (white circle). (**f**) Photograph at the end of suturing after adenomyomectomy.

**Figure 3 jcm-11-06707-f003:**
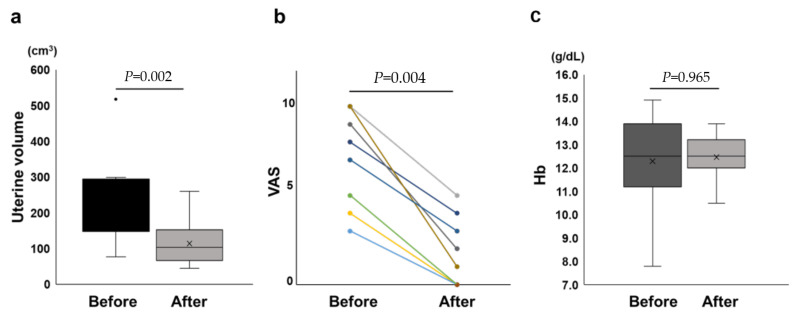
(**a**) Graph of the uterine volumes before and after laparoscopic adenomyomectomy; (**b**) graph of the visual analog scale (VAS) scores before and after laparoscopic adenomyomectomy; (**c**) graph of the hemoglobin levels before and after laparoscopic adenomyomectomy.

**Table 1 jcm-11-06707-t001:** Patients’ preoperative characteristics.

Case	Age (Years)	BMI (kg/cm^2^)	Gravida-Parity	Chief Complaint	Location of Adenomyotic Lesion	Uterine Volume (cm^3^)	Adenomyotic Lesion Volume (cm^3^)	Hb (g/dL)	VAS	Preoperative HT
Case 1	32	26.8	1-0	Hypermenorrhea	Anterior	518	155	10.5	10	Yes
Case 2	39	19.6	1-0	Hypermenorrhea	Posterior	295	137	7.8	8	No
Case 3	35	20.3	0	Dysmenorrhea	Posterior	78	13	14.6	10	Yes
Case 4	37	25.6	0	Hypermenorrhea	Posterior	154	62	14.9	4	Yes
Case 5	41	24.8	0	Dysmenorrhea	Anterior	257	58	11.2	3	Yes
Case 6	40	21.0	1-0	Hypermenorrhea	Anterior	149	29	11.2	5	Yes
Case 7	42	28.1	0	Dysmenorrhea, hypermenorrhea	Anterior	224	11	13.9	8	Yes
Case 8	31	22.0	0	Dysmenorrhea	Posterior	88	14	12.5	7	Yes
Case 9	33	20.7	0	Dysmenorrhea	Posterior	167	19	12.4	9	Yes
Case 10	32	17.5	0	Dysmenorrhea	Posterior	239	90	12.6	10	Yes
Case 11	35	21.3	0	Wish to have a child	Posterior	300	101	13.6	7	No

BMI, body mass index; Hb, hemoglobin; VAS, visual analogue scale, HT, hormonal treatment.

**Table 2 jcm-11-06707-t002:** Operative outcomes.

Case	Operation	Concurrent Operation	Operative Time (min)	Operative Bleeding (mL)	Resected Weight (g)	Uterine Volume (cm^3^)	Hb (g/dL)	VAS (6 Months Later)	VAS (12 Months Later)	Postoperative HT	Follow-up Period (Months)
Case 1	LAM	None	127	450	180.0	261	13.9	2	2	No	30
Case 2	LAM	None	115	300	64.0	154	10.5	3	5	No	14
Case 3	LAM	LM+EDE	125	150	1.5	56	13.0	5	2	Yes	18
Case 4	LAM	None	96	10	50.0	79	13.2	0	0	Yes	14
Case 5	LAM	None	125	150	50.0	152	12.5	0	0	Yes	12
Case 6	LAM	None	88	50	28.5	NA	12.1	0	0	Yes	16
Case 7	LAM	LM	122	150	17.0	103	13.5	4	0	Yes	15
Case 8	LAM	LC+EDE	147	10	10.0	45	11.1	0	NA	Yes	11
Case 9	LAM	EDE	111	50	8.5	71	12.1	2	NA	Yes	9
Case 10	LAM	LM	135	300	68.0	121	12.0	1	NA	Yes	8
Case 11	LAM	LM+LC+EDE	188	100	102.5	104	13.2	3	NA	Yes	7

LAM, laparoscopic adenomyomectomy; LM, laparoscopic myomectomy; EDE, excision of deep endometriosis; LC, laparoscopic cystectomy; Hb, hemoglobin; VAS, visual analogue scale; HT, hormonal treatment; NA, not available.

**Table 3 jcm-11-06707-t003:** Changes in uterine volume, Hb value, and VAS score before and after surgery.

Case	Uterine Volume (cm^3^)			Hb (g/dL)			VAS		
	Before	After	Percentage Change (%)	Before	After	Percentage of Change (%)	Before	Six Months Later	Percentage of Change (%)
Case 1	518	261	−50	10.5	13.9	32	10	2	−80
Case 2	295	154	−48	7.8	10.5	35	8	3	−63
Case 3	78	56	−28	14.6	13	−11	10	5	−50
Case 4	154	79	−49	14.9	13.2	−11	4	0	−100
Case 5	257	152	−41	11.2	12.5	12	3	0	−100
Case 6	149	NA	NA	11.2	12.1	8	5	0	−100
Case 7	224	103	−54	13.9	13.5	−3	8	4	−50
Case 8	88	45	−49	12.5	11.1	−11	7	0	−100
Case 9	167	71	−57	12.4	12.1	−2	9	2	−78
Case 10	239	121	−49	12.6	12	−5	10	1	−90
Case 11	300	104	−65	13.6	13.2	−3	7	3	−57

Hb, hemoglobin; VAS, visual analogue scale; NA, not available.

## Data Availability

The data that support the findings of this study are available from the corresponding author, Y.O., upon reasonable request.

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
