# Peer review of "Laparoscopic Adenomyomectomy under Real-Time Intraoperative Ultrasound Elastography Guidance: A Case Series and Feasibility Study"

_jcm, 2022, doi:10.3390/jcm11226707_

Round 1

Reviewer 1 Report

 In its current form, the manuscript focuses mostly the feasibility of the technique, as part of proof-of-concept study (as indicated on line 248). No comparison is made vs. another technique. No control group is present. Number of patients is low. There is no exclusion criterion. Adenomyosis types are not discussed.

Major comments:

1. The type of lesion should be specified (diffuse, sclerotic, nodular, and cystic) with respect to the methods.

2.  Line 85: please indicate here the % of patient under dienogest at surgery. In table 1, clarify the number of patients on dienogest in post-op. Please consider the effect of HT on the study (see PMID: 29845696), including on stiffness.

3. Please discuss variability in follow-up and why not better standardized. This is indeed a short follow-up (line 246). 

4. Adenomyosis often coexists with uterine fibroids and endometriosis, suggesting a pathological link. Please consider myomectomy or cauterization as a separate variable. Furthermore, reconsider the variable for calculation of average operative time, blood loss.

5. At a minimum, an illustration of clinical ultrasound elastography as guidance imaging (e.g., during surgery) should be presented. Please discuss the type of lesions (such as in Major comment 2) regarding elastography. Please elaborate on the heterogeneity of stiffness between lesions.

Minor Comments:

Discuss the scalability of the technique (learning curve? technology required at other institutions?).

A comment should be made on the development of drug therapies for the management of adenomyosis, as part of background and perspectives. Of note, some of these therapies aim to soften the lesions by reducing the amount of collagen. Please discuss accordingly.

Please discuss possible inclusion/exclusions criteria for future research.

Author Response

Please see the attachment with response letter.

Reviewer 2 Report

This paper evaluates the benefits of intraoperative US-Elastography for the detection of adenomyoma remnants during resection of mild to moderate adenomyosis on eleven patients. 

As the evidence and clinical practice of laparoscopic resection of symptomatic adenomyosis remains unsatisfactory and no common strategy for resection has been broadly accepted the originality and clinical relevance of this work is high.

The English is fine. The graphic presentation is good and the intraoperative pictures of high resolution. For me as someone who has never seen the US-Elastography probe, a picture of the probe would be interesting.

Introduction:

The introduction appears short and conservative pharmaceutical treatment options for adenomyosis are not mentioned (eg. LNR-IUP, gestagen monotherapy, GnRH with or without add back). Moreover, the controversies of the different resection techniques and the need for progress could be outlined.

Methods:

What was the rational for not recruiting a control group where adenomyoma resection was performed without elastography?

The need for analysis of pregnancy rates in patients after adenomyosis-resection is high. It was not mentioned why it was not analyzed.  

Discussion:

The limitations of the study and the related bias could be better outlined. eg. only one surgeon performing the surgeries. Only small to medium adenomyosis size lesions were selected. Again, no control arm was recruited and analyzed. Because of the small collective, and with only one patient wishing to conceive, pregnancy rates were not analyzed.

Round 2

Reviewer 1 Report

The authors have addressed all the comments satisfactorily. The surgical illustration is a good addition to support the manuscript.

Please, review the English language of the amended sentences.